# Reframing Myalgic Encephalomyelitis/Chronic Fatigue Syndrome (ME/CFS): Biological Basis of Disease and Recommendations for Supporting Patients

**DOI:** 10.3390/healthcare13151917

**Published:** 2025-08-05

**Authors:** Priya Agarwal, Kenneth J. Friedman

**Affiliations:** 1Rutgers Robert Wood Johnson Medical School, Rutgers University, New Brunswick, NJ 08901, USA; pa399@rwjms.rutgers.edu; 2Department of Medicine, Rowan-Virtua School of Osteopathic Medicine, Stratford, NJ 08084, USA

**Keywords:** myalgic encephalomyelitis, chronic fatigue syndrome, chronic illness, patient-centered care, advocacy

## Abstract

Myalgic Encephalomyelitis/Chronic Fatigue Syndrome (ME/CFS) is a worldwide challenge. There are an estimated 17–24 million patients worldwide, with an estimated 60 percent or more who have not been diagnosed. Without a known cure, no specific curative medication, disability lasting years to being life-long, and disagreement among healthcare providers as to how to most appropriately treat these patients, ME/CFS patients are in need of assistance. Appropriate healthcare provider education would increase the percentage of patients diagnosed and treated; however, in-school healthcare provider education is limited. To address the latter issue, the New Jersey Myalgic Encephalomyelitis/Chronic Fatigue Syndrome Association (NJME/CFSA) has developed an independent, incentive-driven, learning program for students of the health professions. NJME/CFSA offers a yearly scholarship program in which applicants write a scholarly paper on an ME/CFS-related topic. The efficacy of the program is demonstrated by the 2024–2025 first place scholarship winner’s essay, which addresses the biological basis of ME/CFS and how the healthcare provider can improve the quality of life of ME/CFS patients. For the reader, the essay provides an update on what is known regarding the biological underpinnings of ME/CFS, as well as a medical student’s perspective as to how the clinician can provide care and support for ME/CFS patients. The original essay has been slightly modified to demonstrate that ME/CFS is a worldwide problem and for publication.

## 1. Background

Myalgic Encephalomyelitis/Chronic Fatigue Syndrome (ME/CFS) represents a significant worldwide healthcare challenge due to its debilitating effect on patients and the global lack of understanding of its causes and treatment. The CDC estimates that the worldwide number of individuals afflicted with ME/CFS is somewhere between 17 and 24 million [1]. While data concerning the number of patients with ME/CFS who have been properly diagnosed worldwide is scarce, the weighted average of are cent European survey suggests that an average of 60 percent of ME/CFS patients remain undiagnosed across Europe [2]. In the United States, ME/CFS is estimated to impact 836,000 to 2.5 million people with fewer than 20% of patients having received a formal diagnosis [3,4]. Thus, despite recent advancements in our understanding of the biological basis of ME/CFS, the disease remains misunderstood, with many healthcare providers attributing its origins to psychological causes, further adding to the challenges faced by patients. Educating healthcare providers, who could then provide tailored, supportive therapies and self-help recommendations to their patients, as well as advocate for the reduction in systemic barriers to care, would greatly improve the quality of life for ME/CFS patients.

The New Jersey Myalgic Encephalomyelitis/Chronic Fatigue Syndrome Association is working to raise healthcare-provider awareness about ME/CFS and to combat the stigma still associated with the disease. The organization actively recruits students of the healthcare professions to teach themselves about ME/CFS by offering a guided, independent study: students of the health professions who wish to become the organization’s Medical Scholar Of The Year are asked to write a scholarly paper on a specific topic concerning ME/CFS and the author of the essay judged to be the most scholarly receives a partial tuition remission. The 2024–2025 question to be addressed was to review the evidence supporting the pathophysiological basis of ME/CFS and to suggest methods by which healthcare providers can support their patients’ efforts in improving the quality of their lives. The essay below was written in response to that question, with the hope of raising greater awareness of this global healthcare challenge. Minor changes have been made for publication and in accordance with peer reviewer recommendations.

## 2. Introduction

Myalgic Encephalomyelitis/Chronic Fatigue Syndrome (ME/CFS), formerly referred to as chronic fatigue syndrome (CFS), is a chronic, debilitating condition characterized by profound fatigue that does not improve with rest. Aside from extreme exhaustion, individuals with ME/CFS may experience a range of other symptoms, including post-exertional malaise, cognitive difficulties (commonly called “brain fog”), muscle and joint pain, sleep disturbances, heightened sensitivity to stimuli, flu-like symptoms, and tremors [3]. Other frequently encountered symptoms are lightheadedness and difficulty remaining upright without provoking symptoms [5]. In addition, decreased cerebral blood flow may contribute to symptoms in as many as 90 percent of adults with ME/CFS [6]. Mental health conditions such as depression and anxiety are also common, and ME/CFS patients notably have a greater risk of suicide compared to the general population [3,7].

There is wide overlap of the symptoms found in ME/CFS with those of some other chronic illnesses triggered by infection confounding the diagnosis of ME/CFS. Post-treatment Lyme disease (also known as chronic Lyme disease) and Long COVID both exhibit symptoms with profound overlap of symptoms to those of ME/CFS. The differences between ME/CFS and co-morbid conditions have been reviewed [8]. Currently, there are no biomarkers or clinical tests for ME/CFS and; therefore, its diagnosis is a diagnosis of exclusion as indicated in the Institute of Medicine report of 2015 [5]. The healthcare provider is, therefore, obligated to rule out the diagnoses of both post-treatment Lyme disease, and Long COVID. Our current understanding of the pathophysiological mechanisms producing Lyme disease and the subsequent chronic Lyme disease as well as our current understanding of the pathophysiological mechanisms involved in Long COVID have been recently reviewed in other analyses [9,10].

The United States has produced the highest number of ME/CFS research publications, approximately 8000, with the United Kingdom having produced under 6000 articles. The United States has produced 37 of the most frequently cited top 100 articles, whereas the United Kingdom has produced 32. For this reason, this paper focuses mainly on data derived from U.S. studies. Although ME/CFS impacts an estimated 836,000 to 2.5 million people in the United States, fewer than 20% of patients have received a formal diagnosis from a healthcare provider [3,4].

An additional argument for more aggressive treatment of ME/CFS and its impression is the economic impact of the disease. To our knowledge, the worldwide economic impact of the disease has not been calculated. National Health and Nutrition Examination Survey (NHANES)data from 2021 to 2022 suggest that 1.3% of U.S. adults may have contracted ME/CFS with an ensuing economic cost of approximating $51 billion annually [11]. A second estimate of total annual direct and indirect cost of ME/CFS is $24 billion in the United States [12]. An ever-increasing number of long COVID patients, with many symptoms in common with ME/CFS, provide additional incentives for healthcare providers to learn how to treat these symptoms and manage these patients.

Women are disproportionately affected, with some studies reporting a female-to-male ratio as high as 4:1 [3,4]. The burden of ME/CFS is immense and significantly impacts patients’ quality of life, often leaving them unable to work, pursue education, or engage in activities that previously brought them joy. It is estimated that only about 5% of ME/CFS patients are able to achieve a full recovery and return to their prior state of health [13,14].

ME/CFS patients often face stigma in the healthcare world. Many healthcare providers and researchers have historically dismissed ME/CFS as psychological, or met patients with a lack of understanding and/or disbelief [15]. This mindset invalidated, and in some cases continues to invalidate, the struggle of patients, while also delaying the development of effective treatments. Recent advances in research, however, have uncovered various biochemical and physiological abnormalities associated with ME/CFS, challenging these notions [3].

In addition to the physical challenges, the mental health burden on patients is pro- found. Stigma, lack of understanding, and limited treatment options only further exacerbate the challenges faced by ME/CFS patients. Here we review the literature supporting the biological basis of ME/CFS and provide actionable recommendations for healthcare providers which will alleviate the physical and mental health burdens faced by patients. In so doing, we hope to draw awareness to ME/CFS and promote compassionate, effective care for affected patients.

## 3. Part 1: Challenging Psychological/Psychiatric Causation of Disease

### 3.1. Overview of Biological and Physiological Characteristics of Disease

Currently, ME/CFS does not have a definitive confirmatory test, and is therefore largely a diagnosis of exclusion, often subjecting patients to months and years of symptoms without diagnosis or treatment [13]. Recent research has highlighted the biological basis of ME/CFS, offering avenues for timely diagnosis and development of therapeutics. These advances have helped to validate patient experiences, reduce stigma, and offer hope for earlier diagnosis and development of targeted therapies. These findings also serve to challenge outdated perceptions of ME/CFS as a psychosomatic condition by highlighting its complex, multi-systemic nature. A sampling of some of these recent research findings which underscore the biological and pathological basis of ME/CFS are provided in Appendix A.

ME/CFS onset often follows viral infection, such as COVID-19 or Epstein–Barr virus, though some patients may not have a defined triggering event [13]. The disease has been demonstrated to have a genetic susceptibility, with multiple twin concordance studies demonstrating higher rates of disease in monozygotic twins than dizygotic twins, indicating that certain patients may be at higher risk of disease development depending on their family history [13,16,17,18]. Specifically, three genes, including *IL8*, *NFKBIA* and *TNFAIP3*, have been identified as upregulated in ME/CFS patients [19]. All three of these genes are related to inflammation, circadian clock function, metabolic dysregulation, cellular stress responses, and mitochondrial function, implying that their upregulation could be contributing to the symptomatology of ME/CFS through the disruption of normal physiologic processes [19]. The possibility that altered gene function is associated with ME/CFS is also supported by the frequent co-occurrence of Ehlers-Danlos Syndrome (EDS) and ME/CFS, particularly the joint hypermobility subtype [20,21]. While further studies with larger cohorts of patients are required to confirm these patterns, the developing understanding of potentially altered genes and/or gene expression in ME/CFS patients can begin to pave the way for future research into targeted therapy, such as the use of gene editing.

Immunologic biomarkers of ME/CFS have also been identified. Patients are noted to have increased levels of inflammatory mediators such as pro-inflammatory cytokines, which likely contribute to fatigue and autonomic symptoms [22]. Additionally, ME/CFS patients have been found to have a reduced T-helper (T_H_)1 response, with an overactive T_H_2 response, as indicated by increased levels of interleukin(IL)-10 and a decreased interferon (IFN)-γ/IL-10 ratio, resulting in suppressed cellular immune function with persistent low-grade inflammation [22,23,24]. Therefore, a skewed T_H_1/T_H_2 balance, and the presence of elevated pro-inflammatory mediators such as TNF-α, IL-1, PMN-elastase, lysozyme, and serum neopterin, may point toward a diagnosis of ME/CFS with consideration of the patient’s presentation and clinical picture [22]. Recently, biologics such as monoclonal antibodies are being leveraged in the treatment of chronic illnesses, such as systemic lupus erythematosus and inflammatory bowel disease. As research continues to emerge about immune dysregulation in ME/CFS patients, similar treatment strategies may be able to be explored, though further research is required to confirm safety and efficacy.

Furthermore, neuroanatomical and physiological abnormalities have been noted in ME/CFS patients. Magnetic Resonance Imaging (MRI) has found reductions in the volume of both brain white and gray matter in ME/CFS patients, with the degree of reduction correlating to symptom severity in some patients [22,25,26]. These findings, however, are not necessarily specific to ME/CFS and therefore warrant further exploration. Some patients have also been found to have impairments in brain blood perfusion, as determined by Positron emission tomography (PET); however, these findings are not universal among all patients [22,27]. Furthermore, episodic alterations in brain blood perfusion is demonstrated by an approximate four-fold decrease in internal carotid and vertebral artery blood flow upon patients going from supine to head tilt in comparison to healthy controls using extra cranial Doppler [6]. Blood oxygen level dependent functional MRI (BOLD fMRI) and electroencephalogram (EEG), have uncovered functional differences between ME/CFS patients as compared to healthy controls, with noted increases in brain activity and disrupted brain waves during sleep, further suggesting neurological abnormalities may be contributing to the progression of disease [22,28,29]. These neurological changes could be leveraged by providers as part of diagnostic criteria for ME/CFS, along with other supporting factors from the patient’s history and presentation.

These genetic, immunologic, and neurologic findings highlight the multi-factorial nature of ME/CFS. These discoveries offer avenues for better diagnostic criteria and, therefore, earlier diagnosis. Given how much is still unknown, there is an urgent need for continued research so that targeted therapies can be developed that address the underlying biological mechanisms of ME/CFS.

### 3.2. Countering the Psychological Cause Theory

Though ME/CFS has been classified as a neurological disease by the World Health Organization (WHO) since 1969, many people, including many healthcare providers, still believe it to be a psychosomatic condition [30]. This mindset is ill-informed and neglects decades of research that demonstrate the biological basis of this disease with immunologic, hematologic, and neurologic dysfunction. Additionally, a prospective study of individuals with mononucleosis found that those who progressed to develop ME/CFS exhibited greater physical symptoms and immune abnormalities, but no increase in psychological symptoms, compared to those who recovered [31].

Attributing poorly understood diseases to psychological causes is neither a new nor uncommon occurrence within the medical community. This has been the case for diseases such as rheumatoid arthritis, asthma, fibromyalgia, irritable bowel syndrome, and endometriosis [30]. This historical pattern is notably pronounced for diseases that disproportionately affect women, as seen in the case of ME/CFS, given that the concerns of women are frequently dismissed [30].

## 4. Part 2: Recommendations for Healthcare Providers to Reduce Stigma

### 4.1. Mental Health Burden of Disease

The profound physical fatigue and associated symptoms of ME/CFS lead to a significant mental health burden for affected patients. One study of 169 patients found that a majority (90.5%) felt that there was a lack of understanding of their disease, prompting them to feel uncomfortable sharing their feelings about their condition due to disbelief and trivialization from others [32]. This study also found that 88.2% of patients reported mental distress due to their condition, leading to sadness (71%), hopelessness (66.9%), and thoughts of suicide (39.3%) [32]. The main factors associated with negative mental health were the lack of a cure, feelings of social isolation, and functional limitations [32]. Suicidal thoughts in particular were prompted by being told their condition was psychological (89.5%), lacking strength (80.7%), and feeling misunderstood (80.7%) [32]. These findings demonstrate that ME/CFS is a major public health concern that deserves attention and discussion, in order to promote patient health and well-being. Providers must be aware of both the physical and mental burden of ME/CFS in order to provide effective and compassionate care, particularly with consideration that many patients feel misunderstood or dismissed by their physicians. Such an experience can be isolating and therefore requires nuanced care.

### 4.2. Recommendations for Providers

In order for providers to effectively care for patients with ME/CFS, there must first and foremost be adequate training and education programs in place that emphasize the biological basis of this disease. The U.S. Centers for Disease Control and Prevention (CDC) reports that ME/CFS is not a part of most medical school curricula, unlike other chronic diseases [33]. Therefore, many healthcare providers may be unaware of this disease and therefore may be ill-equipped to care for ME/CFS patients. Early introduction of ME/CFS as a part of the pre-clerkship curricula of medical, physician assistant, and nursing schools would be ideal and would greatly raise provider awareness of the condition, potentially reducing misdiagnoses of patients and connecting them with resources early-on in their disease course. Changing medical school curricula, however, can be challenging and time-consuming. Therefore, a more manageable action item would be to incorporate education about ME/CFS as continuing medical education (CME) lectures throughout residency and beyond, so that current and future physicians are equipped to recognize the condition and remain up to date regarding the newest developments. Education should emphasize research supporting the biological basis of disease, as well as emphasize the mental health burden that patients may face so that providers can screen for troublesome signs such as suicidality.

When it comes to caring for patients, providers must offer compassionate, evidence-based counseling, with recommendations for helpful resources for symptom management, education, and mental healthcare. Providers need to be aware that ME/CFS patients generally require longer office visits than most other patients. An analysis of U.S. healthcare data found that ME/CFS-related appointments lasted ~23.6 min in comparison to ~19.4 min for other visits [34]. Providers may wish to consider end-of-day appointments for ME/CFS patients to ensure sufficient time. While reimbursement mechanisms that support longer, more resource-intensive provider visits for patients with complex, chronic illnesses are beginning to appear (e.g., HCPCS Add-On Code G2211- Add-on for complex longitudinal visits, CPT 99490/99439-Monthly non-face-to-face coordination, and 99487/99489-Complex CCM), healthcare providers should be encouraged to advocate for adequate compensation by influencing federal and payer-level reimbursement policies. Traditional, professional avenues of advocacy for change in healthcare are through national and specialty-specific associations, contributing to position papers, letters of comment, and testimonials to both CMS and Congress, and partnering with disease-specific (in this case ME/CFS-specific) advocacy organizations whose missions include raising awareness among lawmakers and insurers about the unmet healthcare needs of their constituency. More healthcare providers might be willing to accept patients with complex, chronic illness were the time required to manage these patients adequately compensated.

Providers can encourage patient self-education by directing patients to the CDC’s ME/CFS page, which offers comprehensive information on symptoms, management strategies, and printable handouts [35].

Similarly, the Solve ME/CFS Initiative is a great resource for patients that provides webinars, patient stories, and updates on current research. Additionally, providers can recommend participation in support groups, such as those hosted by the ME Action Network or other online communities. Patients may find strength in connecting with others who share their experiences, reducing feelings of isolation and stigma.

In terms of symptom management, physicians should introduce exertional pacing strategies that teach patients how to manage their limited energy to avoid post-exertional malaise [36]. Apps such as Visible can also help patients identify triggers, manage energy levels, and monitor their progress overtime [37]. Additionally, providers should ensure that patients are regularly screened for mental health concerns. Physicians can specifically recommend therapists with experience in chronic illness and recommend simple tools for mindfulness, such as the Calm or the Headspace app, to support stress management and improve sleep quality [38,39].

Additionally, providers can suggest practical resources to improve patients’ activities of daily living, such as mobility assistive devices, home modifications (e.g., in-shower benches, stair lifts, etc.), and/or physical therapy. Some patients may also benefit from dietary counseling by a dietitian, with recommendations for foods and supplements that target inflammation and optimize energy levels. Finally, patients should be informed about opportunities to participate in ongoing research/clinical trials should they be available through local organizations, as these may afford them access to innovative treatments while also offering an opportunity to directly participate in enhancing the world’s understanding of ME/CFS. By offering numerous avenues for support, providers can empower patients and improve their quality of life as they navigate life with ME/CFS.

With recognition of the barriers to care that ME/CFS patients may face, providers should also strive to ensure equitable and accessible care for all. Whenever possible, physicians should offer opportunities for telemedicine visits, particularly for patients who may find it difficult to come to an office in person due to their symptoms. Offices should be accessibly designed, so that patients who may require assistive mobility devices can navigate without difficulty, as well. Taking steps such as these reduces potential barriers to care and helps ensure that patients are receiving the healthcare they deserve.

Ultimately, healthcare providers play a big role in the journey of an ME/CFS patient. Through education, tailored supportive recommendations for patients and reduction in systemic barriers to care, physicians can greatly improve the quality of life for patients experiencing ME/CFS.

Physicians should prioritize genuinely listening to their patients’ concerns, fostering trust and understanding. While physicians may not have all the answers, as much remains unknown about ME/CFS, it is essential for physicians to make every effort to connect patients with appropriate resources and support that can address their needs.

## 5. Conclusions

ME/CFS is a multi-system, chronic disease that profoundly impacts the daily lives of patients. Despite significant research advancements in understanding the biological basis of ME/CFS, it has long been misunderstood, with many attributing its origins to psychological causes, further adding to the challenges faced by patients. ME/CFS imparts a significant mental health burden on patients, leading to increased rates of depression, anxiety, and suicide [7].

Studies have demonstrated that patients with ME/CFS have alterations in genes associated with inflammation, metabolic regulation, and mitochondrial function, alongside irregularities in immune pathways marked by increased pro-inflammatory activity [13,19,22]. Neuroimaging studies have further demonstrated reductions in white and gray matter, as well as decreased brain perfusion [22]. These studies taken together illustrate the cascade of biological alterations in ME/CFS patients, strongly contradicting the notion that ME/CFS is merely a psychological disorder.

With recognition of the potentially devastating impact ME/CFS can have on the lives of patients and their families, healthcare providers have an obligation to be well-prepared to provide effective and compassionate care. Medical education and continuing professional development must prioritize teaching the biological underpinnings of ME/CFS, along with effective strategies to support patients. Providers can support patients by offering a plethora of resources including support groups, mental health services, and mobility aids. Physicians must also make efforts to reduce barriers to care, such as by offering telemedicine visits and accessible offices. Through these strategies, physicians can cultivate trust and improve the quality of life for patients with ME/CFS.

Ultimately, by addressing the biological complexities of ME/CFS, challenging outdated misconceptions, and fostering a supportive healthcare environment, we can pave the way for better outcomes and a brighter future for those living with ME/CFS.

## Data Availability

Any additional inquiries concerning the data presented in this manuscript may be directed to K.J.F., the corresponding author.

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
