# Peer review of "Reframing Myalgic Encephalomyelitis/Chronic Fatigue Syndrome (ME/CFS): Biological Basis of Disease and Recommendations for Supporting Patients"

_healthcare, 2025, doi:10.3390/healthcare13151917_

Round 1
Reviewer 1 Report
Comments and Suggestions for Authors
This essay is the work of a student who was tasked with reviewing the evidence supporting the pathophysiologic basis of ME/CFS. It is a very nice summary of evidence against the psychosomatic basis of the illness, highlighting a number of the objective abnormalities that have been discovered. I had a few suggestions for improving the scope of the physiologic disorders being discussed, very much consistent with the author's message. These are abnormalities that contribute to the pathophysiology of ME/CFS, are not under conscious control, and therefore support the notion of a physiologic contribution to the illness.
Specific comments:
Background, line 7: omit the comma after "a recent European survey"
Page 2, first paragraph of the Introduction: a major omission from the symptoms is lightheadedness and difficulty remaining upright without provocation of symptoms. Orthostatic intolerance is highlighted in the Institue of Medicine case definition for ME/CFS, and is seen in over 95% of pediatric ME/CFS patients (see the IOM book on ME/CFS and also Roma M, et al. Impaired health-related quality of life in adolescent myalgic encephalomyelitis/chronic fatigue syndrome: the impact of core symptoms. Frontiers in Pediatrics 2019 (Feb 15);7:26). Recent work shows that it is also present in 90% of adults (see van Campen CMC, et al. Cerebral blood flow is reduced in ME/CFS during head-up tilt testing even in the absence of hypotension or tachycardia: a quantitative, controlled study using Doppler echography. Clinical Neurophysiology Practice 2020; 5:50-58).
Introduction, paragraph 2: a good reference to include and discuss is the recent CDC data describing the NHANES data from 2021-2022 showing that 1.3% of US adults had ME/CFS. That paper also describes the economic costs as being up to 51 billion dollars annually, so the authors might want to include the additional references and not just cite the single paper (ref 8) by Jason and colleagues. (NCHS Data Brief No. 488 December 2023, Myalgic Encephalomyelitis/Chronic Fatigue Syndrome in Adults: United States, 2021–2022
Anjel Vahratian, Ph.D., M.P.H., Jin-Mann S. Lin, Ph.D., Jeanne Bertolli, Ph.D., M.P.H., and
Elizabeth R. Unger, Ph.D., M.D.)
These numbers are likely to grow more due to the big influx of post-pandemic patients with ME/CFS.
Part 1, Challenging Psychological /Psychiatric Causation of Disease:
In paragraph 2, when discussing the genetic contributions to disease, the authors might want to highlight the consistent data on joint hypermobility and Ehlers-Danlos syndrome as risk factors for ME/CFS. These patients have objective physical examination abnormalities that also argue against a psychological cause of the illness. As far as I am aware, there is only one study from the Jason group that challenges this notion, and it had a fatal flaw in the measurement of the Beighton score in patients, as well as incomplete assessment of all patients. In contrast, several studies emphasize the higher rates of heritable connective tissue laxity in ME/CFS (
Nijs J, Aerts A, De Meirleir K. Generalized joint hypermobility is more common in chronic fatigue syndrome than in healthy control subjects. J Manipulative Physiol Ther. 2006 Jan;29(1):32-9. doi: 10.1016/j.jmpt.2005.11.004. PMID: 16396727.
In this section, possibly in paragraph 4 where brain blood perfusion is discussed, I would add the work on van Campen and Visser on the reductions in cerebral blood flow observed in adults with ME/CFS. These authors examined 429 patients with ME/CFS and 44 healthy controls, measuring cerebral blood flow using extracranial Dopler of the internal carotid arteries and vertebral arteries, adding the flow though those four vessels to obtain total cerebral blood flow (CBF). In healthy individuals, compared to supine measurement, CBF fell 7% during 30 minutes of head-up tilt to 70 degrees. In ME/CFS patients, the mean reduction was 26%, a remarkable and significant reduction that patients obviously could not manipulate. This work is more compelling than the PET data, and should be highlighted both in this section of hte manuscript and also in Appendix A.
The section on Recommendations for Healthcare providers is ful of helpful suggestions. Do the authors want to include a comment on the fact that funding of health care for those with complex chronic illness needs to be changed to account for the time required to manage complex patients. Listening to patients might be adopted more readily if the time was reimbursed.
Author Response
Comments for Reviewer #1:
We appreciate the suggestions made. We agree that the suggestions would strengthen the manuscript.
Accordingly, we have incorporated the suggestions:
-
The symptoms of lightheadedness and difficulty standing have been added.
-
Orthostatic intolerance and inclusion of the Institute of Medicine report have been added,
-
The larger reduction in cerebral blood flow observed in ME/CFS patients upon going from a supine position to upward head tilt position has been added.
-
The observed decreased cerebral blood flow found in ME/CFS patients as measured by extra-craneal Dopler has been added.
-
Mention of the NHANES data and the economic cost of ME/CFS has been added,
-
The comorbidity of Ehlers-Danlos Syndrome, particularly the joint hypermobility subset in ME/CFS patients has been added.
-
The insufficient compensation of healthcare providers and their difficulty in obtaining adequate compensation for their spending increased office visit time for treating ME/CFS patients has been added.
-
Suggestions as to how healthcare providers can professionally advocate for increased compensation for managing these patients has been added.
These additions should strengthen our paper and satisfy the reviewer.
Reviewer 2 Report
Comments and Suggestions for Authors
Although this article provides a strong perspective in terms of literature presentation and patient support recommendations, it has serious methodological deficiencies in terms of scientific method. First of all, the article does not contain any empirical research and is not based on any structured scientific methods such as original data analysis, systematic review or meta-analysis; this situation reduces the article to the level of an interpretive opinion piece. Since the selection of the sources used was not made with a systematic method, the literature coverage may be biased or incomplete. In addition, the suggested biological mechanisms (e.g. genetic predisposition, immunological disorders, neuroimaging findings) are presented at a very general level, but no concrete methodological roadmap is given on how to integrate these findings into clinical diagnosis and treatment. Again, the suggested support strategies (e.g. exercise tracking applications, online support groups, nutritional counseling) remain at a theoretical level, and no evidence-based analyses are presented on the effectiveness of these strategies.
Author Response
Comments for Reviewer #2:
That our manuscript disappoints the reviewer is unfortunate. However, we state both the intent and the type of our manuscript in the manuscript’s Introduction. We believe our manuscript is of value to healthcare providerswho are considering providing care and management to patients of ME/CFS, Long COVID, and similar chronic diseases triggered by infection. The manuscript will also be of value to patients with any of these conditions who have found healthcare providers willing to consider providing care for such patients but are unfamiliar with the available methods and strategies. Patient may bring copies of this article to such providers (when it is published).
The manuscript presents the arguments for ME/CFS healthcare and how to provide it written by a medical student. Who better to provide a summary of the pathophysiological basis of the disease and the tools currently available for care and self-care?
The manuscript is to be published in the PAPIS Topical Collection Healthcare. As stated on the PAPIS Topical Collection Information, page, a wide range of manuscripts are acceptable to this collection. Se:e: (Healthcare | Topical Collection : Why Some Patients Never Fully Recover: Post Active Phase of Infection Syndromes (PAPIS))
While the reviewer believes our literature review may be biased, our incorporation of the literature suggested by reviewer #1 should assuage or eliminate that fear of bias.
While it is true that the support strategies presented in this paper have not been peer reviewed, it is important to remember that ME/CFS, initially described by the Public Health Service in 1934, has been underfunded and under-researched up to the present time. Comparatively few if any studies concerning the effectiveness of support strategies in palliating the symptoms or improving the lives of ME/CFS patients have been published. However, millions of ME/CFS patients have used these strategies and have reported improvement. Given these circumstances, we believe it is not reasonable to suppress patient reports.
While the reviewer asks us to provide a road map of clinical care for ME/CFS patients, such road maps have been previously published e,g, the IACFS/ME Adult Primer and the International Authoring Committee’s Pediatric Primer. We also caution against a road map or cookbook approach to ME/CFS treatment because the range of presentation of ME/CFS is highly variable. The general rules-of-thumb for providing care to ME/CFS patients are to: 1. Address the patient’s most disturbing symptom first, 2. Address only one symptom at a time, and 3. Start medications (and supplements) at one quarter of the normal dose due to chemical sensitivities in the majority of these patients.
We hope our responses will satisfy this reviewer.
Round 2
Reviewer 1 Report
Comments and Suggestions for Authors
The authors have responded in a thorough manner to the reviewer suggestions.
Author Response
We appreciate the concern of the reviewer and the opportunity to strengthen our manuscript.
Reviewer 2 Report
Comments and Suggestions for Authors
accept
Author Response

(The authors gave the same response as above.)
